# A Modified Modulation Strategy for an Active Neutral-Point-Clamped Five-Level Converter in a 1500 V PV System

**Guodong Chen** [1] **and Jiatao Yang** [2,*]

1   Technology Center, Shanghai Electric Power Transmission & Distribution Group, Shanghai 200042, China; chengd@shanghai-electric.com
2   Key Laboratory of Control of Power Transmission and Conversion, Ministry of Education, Shanghai Jiao Tong University, Shanghai 200240, China
*   Correspondence: sjtu-sd-zzc@sjtu.edu.cn

**Abstract:** With the development of 1500 V photovoltaic (PV) systems in recent decades, multilevel inverters such as the five-level inverter have gained much attention for their higher equivalent output frequency and low semiconductor devices' voltage stress. Among five-level inverters, the active neutral-point-clamped five-level (ANPC-5L) inverter is very competitive due to its simple structure and control methods. However, with its conventional commutation strategy, the topology of the ANPC five-level converter has the security risk of overvoltage in the power device when switching to dead time under special conditions, which affects the reliability and safety of the switch state switching process. In this paper, this issue is analyzed in detail and a modified commutation strategy is proposed. Meanwhile, a novel soft start-up method adopted to an ANPC-5L inverter is also proposed. A prototype is also set up to analyze the issue of traditional switching commutation strategies and to verify the effectiveness of the proposed commutation strategy and the soft start-up method.

**Keywords:** ANPC-5L converter; reliable switching state; modulation; PV grid-tied inverter

## 1. Introduction

Photovoltaic generation has been paid more attention recently because of the shortage of fossil fuels and the increasingly serious levels of environmental pollution, which play an important role in PV systems [1]. Compared with previous 1000 V systems, the 1500 V system reduces the number of cables and PV plants, and decreases the line cost and conduction loss [2,3]. Moreover, it provides more voltage range which is used to ensure maximum power point (MPPT) availability by controlling front-end circuits or adjusting the grid-connected voltage [4,5].

Nowadays, multilevel inverters such as the five-level inverter have gained much attention for their high equivalent switching frequency and low voltage stress, which are benefits for increasing the inverter's power density [6,7]. The neutral-point-clamped (NPC) inverter, flying capacitor (FC) inverter and T-type inverter are traditional three-level inverters which have been widely employed in industrial application. The NPC inverter is generally adopted in centralized PV grid-tied inverters because of its simple operation principle and high power level capability [8,9], which are different from the demands of PV string inverters. When used in low bus voltage applications, the T-type inverter is suitable on account that it can reach a higher work frequency, higher conversion efficiency, and higher power density [10,11]. The unbalance of neutral-point voltage is the main issue in multilevel inverters, except in the FC inverter [12]. However, one more floating capacitor is added in each phase, resulting in a larger volume and poorer power density, and its control scheme is more complex.

Many efforts have been made on topologies for photovoltaic multilevel inverters. Five-level topology reduces both the voltage stress of semiconductor devices and the volume of filter inductance compared with three-level topology due to its better harmonic performance, which may lead to loss reduction and system cost reduction. NPC-5L is the usual topology used for five-level topology [13], in addition to other topologies such as cascaded H-bridge five-level (CHB-5L) and FC five-level (FC-5L). Problems such as relatively large switching losses, unbalanced voltages of capacitors, and poor stability have promoted research into five-level topologies [14]. Other different topologies of multilevel inverters have also been adopted in industrial application: the stacked multi-cell (SMC) [15], the H-bridge NPC (H-NPC) [16], the neutral-point piloted (NPP) [17], and the modular multilevel converter (MMC).

The ANPC-5L converter, as shown in Figure 1, has been paid more and more attention since it was proposed [18,19], and is the combination of two types of inverters. One is the FC three-level inverter, the other is the active NPC three-level inverter. The advantages of this inverter consist of low switching losses and the convenience of capacitor voltage balance [20,21]. Switches which are connected in series in this topology switch at fundamental frequency, while the others switch at carrier frequency. Meanwhile, the switching cost of this topology is low because the stress of the switches is $V_{DC}/4$, while $V_{DC}$ is the voltage of DC-link. Moreover, if different switching states are chosen appropriately, the voltage of floating capacitors is easy to balance.

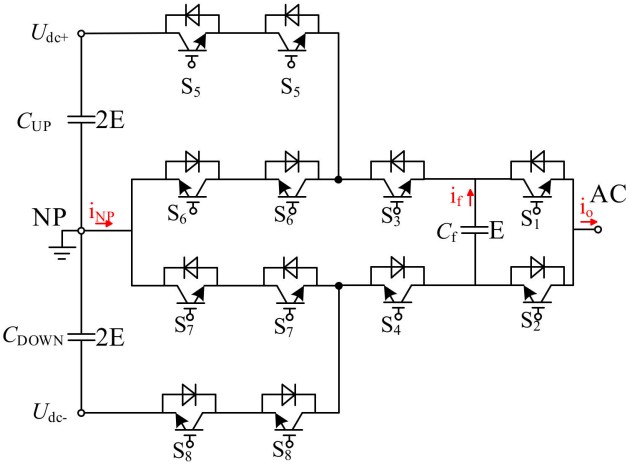

**Figure 1.** Topology of the ANPC-5L inverter.

Researchers have carried out a lot of work on ANPC-5L modulation technology, flying capacitor voltage control, neutral-point voltage control, and other issues [22,23]. The modulation strategy of the ANPC-5L inverter is simple and reliable most of the time. However, less attention has been paid to the voltage stress of switching devices in ANPC-5L converters, and the voltage stress of switching devices is very important for the safe and reliable operation of inverters. Document [24] analyzes the operation state of inverters based on space vector pulse-width modulation, including 125 vector combinations, and limits the stress of the switching devices by using the safe switching state switching process. Meanwhile, under the conventional modulation scheme, the analyzed inverter has the security risk of overvoltage in the power device when switching to dead time at the zero-crossing point of voltage when the output current is inductive, which affects the commutation safety [25].

In this article, this issue is analyzed in detail and a modified modulation strategy is proposed. In comparison with other modulations, this method provides several free degrees which are used to ensure the elimination of the voltage stress of power devices by choosing favorable circuit states and controlling current commutations. Additionally, it can realize the flying capacitor voltage balance in several carrier wave periods. The implementation of

the proposed strategy in digital systems is rather simple. Meanwhile, a novel soft start-up method adopted to the ANPC-5L inverter is also proposed. Experimental results prove that the proposed strategy is valid.

The rest of the paper is organized as follows. Section 2 presents the traditional switching states. Section 3 analyzes the power device's overvoltage issue in detail. Section 4 proposes the modified modulation strategy to solve the potential safety hazards. Section 5 proposes a control method for soft start-up. Section 6 illustrates an inverter prototype for analysis verification. The conclusions are given in Section 7.

## 2. Traditional Modulation Strategy

As shown in Figure 1, the ANPC five-level inverter has eight power switches S1–S8, a floating capacitor $C_F$, a upper capacitor $C_{UP}$, and a lower capacitor $C_{DOWN}$. For the ANPC-5L inverter, as shown in Table 1, the conventional modulation scheme uses eight basic switching states to produce five voltage levels. There are redundant states at the +E and −E levels (E is 1/4 of bus voltage $V_{DC}$) which affect the charging or discharging states of the floating capacitors. The balance of the floating capacitor voltages can be realized by choosing appropriate switching states.

**Table 1.** Traditional switching states of the ANPC-5L inverter.

| State | S1 | S2 | S3 | S4 | S5 | S6 | S7 | S8 | Vo | Vcf Io > 0 | Vcf Io < 0 |
|-------|----|----|----|----|----|----|----|----|------|---------|---------|
| V1 | 0 | 1 | 0 | 1 | 0 | 1 | 0 | 1 | −2E | - | - |
| V2-1 | 0 | 1 | 1 | 0 | 0 | 1 | 0 | 1 | −1E | C | D |
| V3 | 1 | 0 | 0 | 1 | 0 | 1 | 0 | 1 | −1E | D | C |
| V4-1 | 1 | 0 | 1 | 0 | 0 | 1 | 0 | 1 | −0 | - | - |
| V5-1 | 0 | 1 | 0 | 1 | 1 | 0 | 1 | 0 | +0 | - | - |
| V6 | 0 | 1 | 1 | 0 | 1 | 0 | 1 | 0 | +1E | D | C |
| V7-1 | 1 | 0 | 0 | 1 | 1 | 0 | 1 | 0 | +1E | C | D |
| V8 | 1 | 0 | 1 | 0 | 1 | 0 | 1 | 0 | +2E | - | - |

The harmonics of the phase disposition (PD) contain a few carrier harmonics because of the different phases of the four carrier waves, as well as DC components, fundamental components, and carrier sidebands. However, some other modulation schemes, such as alternative phase opposition disposition (APOD), phase opposition disposition (POD), and two kinds of phase-shift carriers (PSC), have no carrier harmonics. In terms of single-phase inverters, the harmonic performance of the above-mentioned modulation is just the same due to the signal–energy conservation law. As far as the three-phase system is concerned, the harmonics are quite different. When the carrier waves of the three-phase system are synchronous, the carrier harmonics of the adjacent phases will exactly coincide, which represents that this harmonic will not appear in line voltage. However, carrier sidebands have no similar features. The harmonic performance of PD is the best. The next is APOD and PSC. The worst is POD. The characteristics of line voltage spectrums are far more diverse in the cause of common mode voltage. Eventually, considering the harmonic performance, the PD-PWM method is preferable among various methods. Under the traditional PD modulation of the ANPC-5L inverter, there are four cascaded carrier waves. As shown in Figure 2, comparison with the first carrier wave makes the output voltage change between +2E and +E. Switching between these states changes only two switch devices, and the switching processes are safe. Similarly, the switching processes in comparison with the other carrier waves are also reliable.

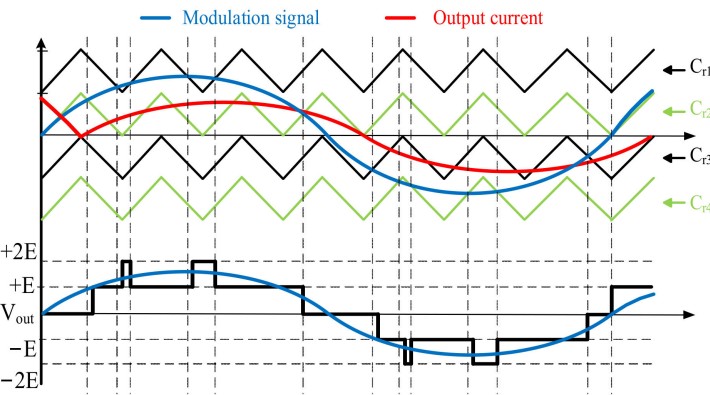

**Figure 2.** Traditional modulation strategy of the ANPC-5L inverter.

## 3. Overvoltage Issue of Traditional Modulation

However, there are potential overvoltage issues in the conventional modulation scheme. As shown in Figure 3, according to the counter mode of up–down or down–up, the output voltage will change from +E to −0 or from +0 to −E. Unlike the former switching process, switching between these two states changes six switch devices. Although the dead time of each pair of devices' switch exists, there will be safety problems under certain circumstances.

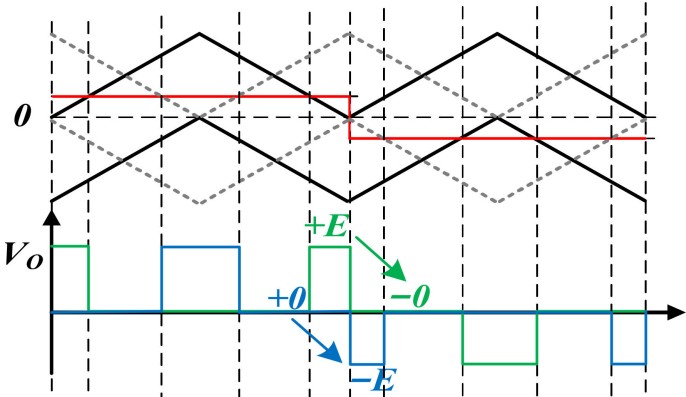

**Figure 3.** Switching process at zero crossing point.

As shown in Figures 4 and 5, taking the change from +0 to −E as an example, the circuit changes from state V5-1 to V2-1. Due to the existence of the dead zone, all the devices are turned off and there will be an intermediate state ($V_{DANGER}$ (S1-S8:01000000)), which will cause the overvoltage issue.

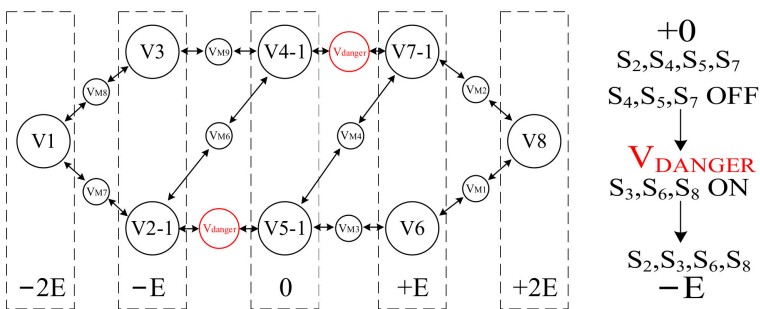

**Figure 4.** Traditional switching process of the ANPC-5L inverter.

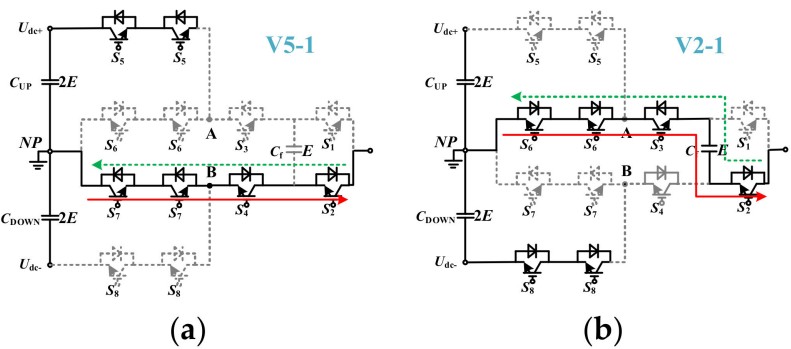

**Figure 5.** Switching states V5-1 and V2-1 at zero crossing point. (**a**) Switching states V5-1. (**b**) Switching states V2-1.

A detailed analysis is presented as below. In the state of V5-1, the initial states of the parasitic capacitance of the MOSFETs are shown in Figure 6; S3 has a potential difference of E while S6 has 2E. During the dead time of S5, assuming the output current $I_O$ is greater than zero, S3 and S4 are all off. The continuous current path is shown in Figure 6, and the final states of the parasitic capacitance decide the safety of the commutation process.

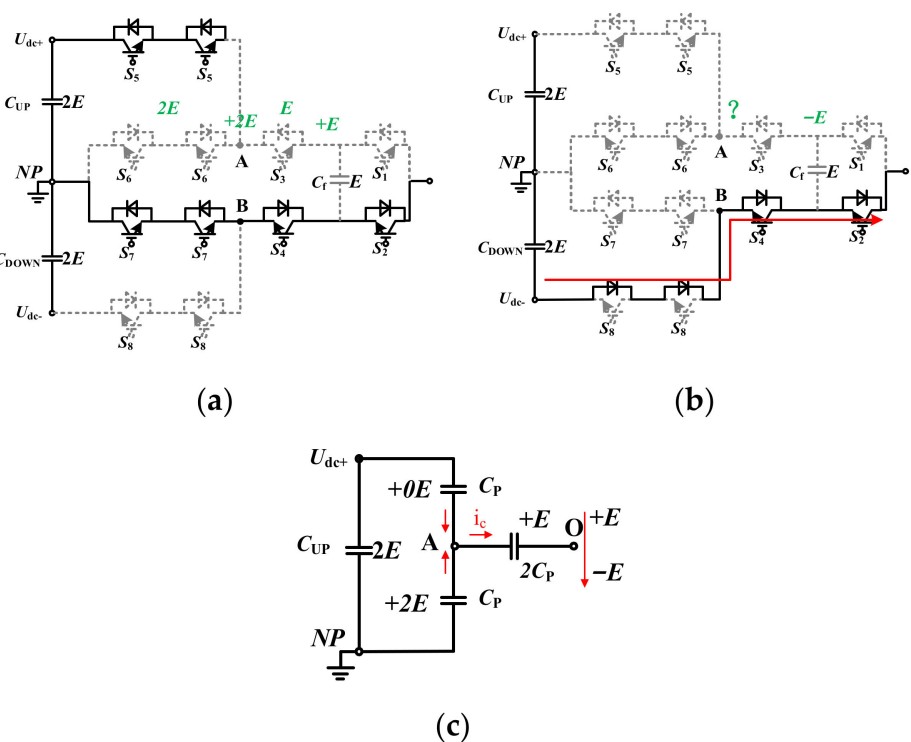

**Figure 6.** (**a**) Initial potential difference of V5-1. (**b**) Potential difference of $V_{DANGER}$. (**c**) Equivalent circuit of the charging process from V5-1 to $V_{DANGER}$.

During the dead time, S3, S5 and S6 are all off, closed switches equivalent to parasitic capacitances. Therefore, as shown in Figure 6, the equivalent circuit of the switching process is equal to the charging of parasitic capacitances.

Because of the charge–balance principle, increased charge on S3 should be equal to the summation of increased charge on S5 and reduced charge on S6. Moreover, the sum of the voltages of the S5 and S6 constants is equal to 2E, and the increased voltage on S5 is equal to the reduced voltage on S6. According to the capacitance definition:

$$dQ = C \cdot dU \tag{1}$$

Equation (2) can be obtained:

$$C_p \cdot dU_{S5} + C_p \cdot dU_{S6} = C_p \cdot dU_{S6} + C_p \cdot dU_{S6} = 2C_p \cdot dU_{S3} \tag{2}$$

The voltage of endpoint O changes from +E to −E by focusing on the steady state of −E, and according to Kirchhoff's Voltage Law (KVL) Equation (3) can be obtained:

$$2E - dU_{S6} = E + dU_{S3} + (-E) \tag{3}$$

Combining the above two formulas, increased voltage on S3 can be obtained:

$$dU_{S3} = E \tag{4}$$

Therefore, as shown in Figure 7, the final voltage of S3 will be 2E. The voltage stress will be higher than the voltage the device can withstand, which may cause overvoltage breakdown and influence the normal operation of the circuit.

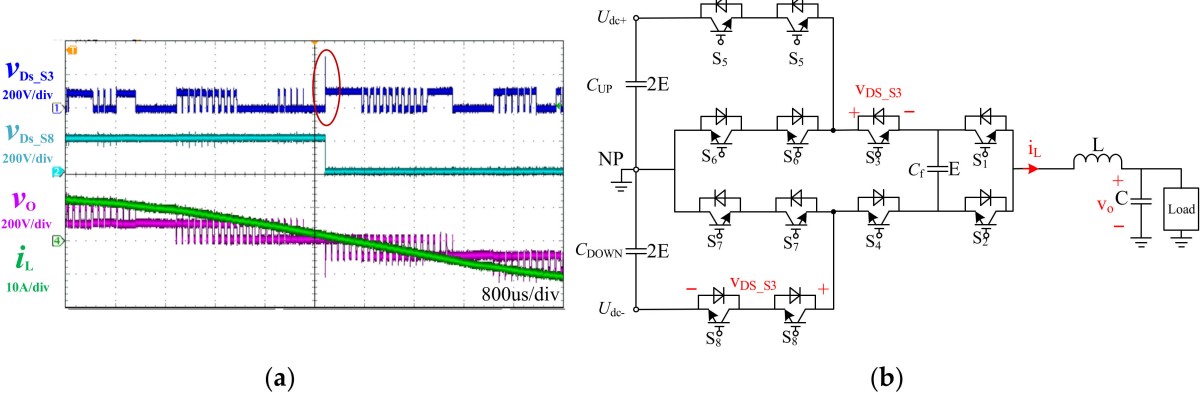

**Figure 7.** The overvoltage stress issue. (**a**) Experimental waveforms. (**b**) Experimental circuit.

Before the switching process of S5, there are three possible states for S3 and S4. As shown in Figure 8, when S4 is on and S3 is off, or S3 and S4 are both off, assuming the output current IO is greater than zero, the conduction path and the charging process are analyzed as above.

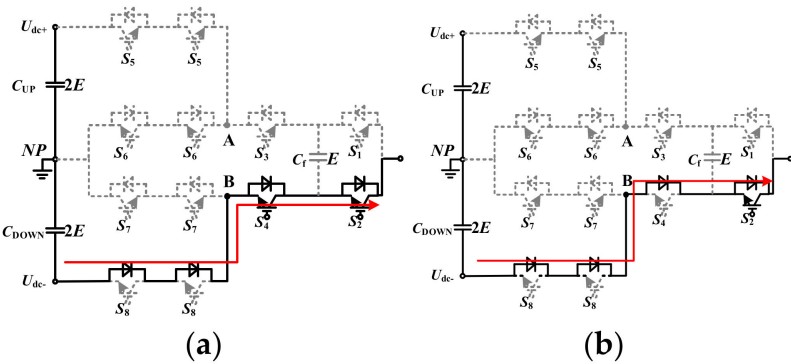

**Figure 8.** Conduction path of S3 and S4 equals to 0.1 or 0.0. (**a**) Positive current. (**b**) Negative current.

However, when S4 is off and S3 is on, as shown in Figure 9, the switching process is different. If the $I_O$ is greater than zero, turning off S5 causes the conduction of S6's reverse diode, the voltage of point O changes from +E to −E and the initial voltage on S8 is 2E. The process seems similar to the above situation, but the direction of S4 decides that the charging current $I_C$ reduces its voltage to zero; then, the reverse diode will conduct and there will be no overvoltage risk. However, the output voltage goes through 0, +E, −E. The

ideal process is 0 to −E, although the commutation is not perfect. Therefore, a modified switching method is proposed to achieve the best commutation.

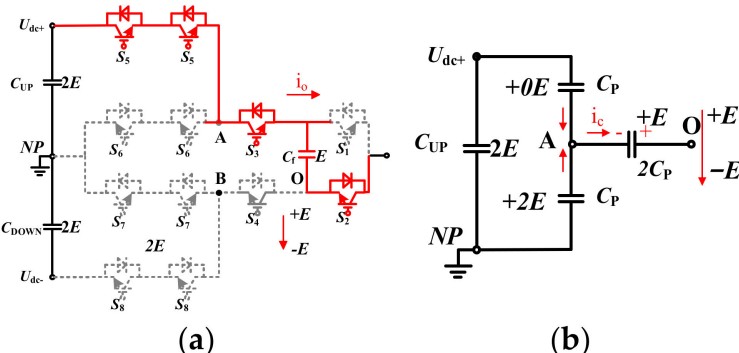

**Figure 9.** (**a**) Conduction path and potential difference when S3, S4 equals to 1.0 (**b**) Equivalent circuit of the charging process.

## 4. Proposed Modified Modulation Strategy

The conventional modulation scheme only considers eight simple states. After analyzing the switching states and overvoltage issues, eight other states (V2-2, V2-3, V4-2, V4-3, V5-2, V5-2, V7-2, V7-3) are obtained, as shown in Table 2, which can be included in the state cutover.

**Table 2.** The ANPC-5L converter's switching states and influence on the voltage of the flying capacitors.

| State | $S_{x1}$–$S_{x8}$ | $V_{cf}$ | | $V_{ox}$ | Level |
|:---:|:---:|:---:|:---:|:---:|:---:|
| | | $I_o > 0$ | $I_o < 0$ | | |
| $V_1$ | 01010101 | – | – | $-V_{dc}/2$ | −2 |
| $V_{2\text{-}1}$ | 01100101 | | | | |
| $V_{2\text{-}2}$ | 01100100 | C [a] | D | $-V_{dc}/4$ | −1 |
| $V_{2\text{-}3}$ | 01110100 | | | | |
| $V_3$ | 10010101 | D | C | $-V_{dc}/4$ | −1 |
| $V_{4\text{-}1}$ | 10100101 | | | | |
| $V_{4\text{-}2}$ | 10100100 | – | – | 0 | 0 |
| $V_{4\text{-}3}$ | 10110100 | | | | |
| $V_{5\text{-}1}$ | 01011010 | | | | |
| $V_{5\text{-}2}$ | 01010010 | – | – | 0 | 0 |
| $V_{5\text{-}3}$ | 01110010 | | | | |
| $V_6$ | 01101010 | D | C | $V_{dc}/4$ | 1 |
| $V_{7\text{-}1}$ | 10011010 | | | | |
| $V_{7\text{-}2}$ | 10010010 | C | D | $V_{dc}/4$ | 1 |
| $V_{7\text{-}3}$ | 10110010 | | | | |
| $V_8$ | 10101010 | – | – | $V_{dc}/2$ | 2 |

[a] C: charging; D: discharging.

By combining additional circuit states, a modified modulation strategy and complete state machine are proposed, as shown in Figure 10. Under the proposed modulation scheme, a safe switching process within a power circle can be achieved and there will be no overvoltage stress, as shown in Figure 11.

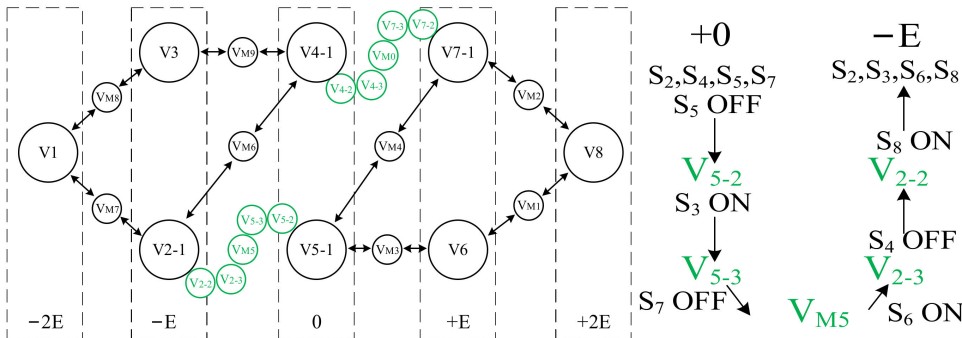

**Figure 10.** Complete state machine for the ANPC-5L process.

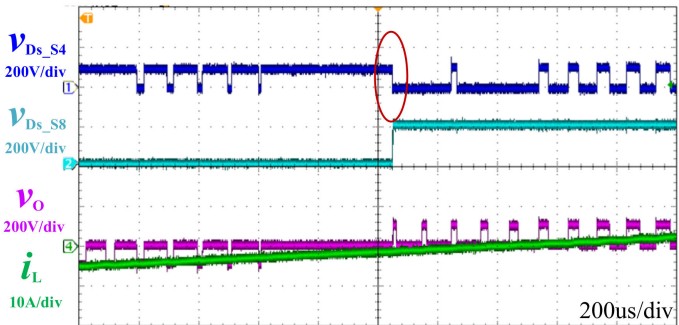

**Figure 11.** Reliable switching process of proposed modified modulation strategy.

The presented modulation strategy for the ANPC-5L converter is deeply researched in this section. Consisting of modulation signal, output current, driving signal and the voltage of the flying capacitors, Figure 4 shows the schematic diagram. With the scheme that is proposed, the switches, which are series-connected in the ANPC-5L converter, switch at fundamental frequency while the others switch at carrier frequency, which results in low switching losses. In the meantime, low switching and conduction loss can be achieved because the stress of all devices can be restricted to Vdc/4.

A.    Flying capacitor voltage balance

The voltage of floating capacitors can be affected by different circuit states. One is E and the other is −E (V6, V7-1 and V2-1, V3), which can cause the capacitor to charge or discharge. To go a step further, we can use +E and −E levels within a sinusoidal voltage wave. If circuit states are selected appropriately, the balance of the voltage of FC will be achieved during the whole cycle.

Figure 12 shows that with a positive output current in V6, the flying capacitor is charging, and with a negative output current in V7-1 the flying capacitor is in a discharging state. With the increasing carrier frequency in each fundamental period, the balance of FC voltage will be better controlled in several carrier periods. Because the carrier wave period is much shorter than the fundamental period, the capacity value of the flying capacitor in the ANPC-5L inverter can be significantly reduced for a definite maximal permissible voltage range, compared with those inverters controlling a fundamental period. Ultimately, with the reduced volume of the flying capacitor, the power density of the inverter increases significantly.

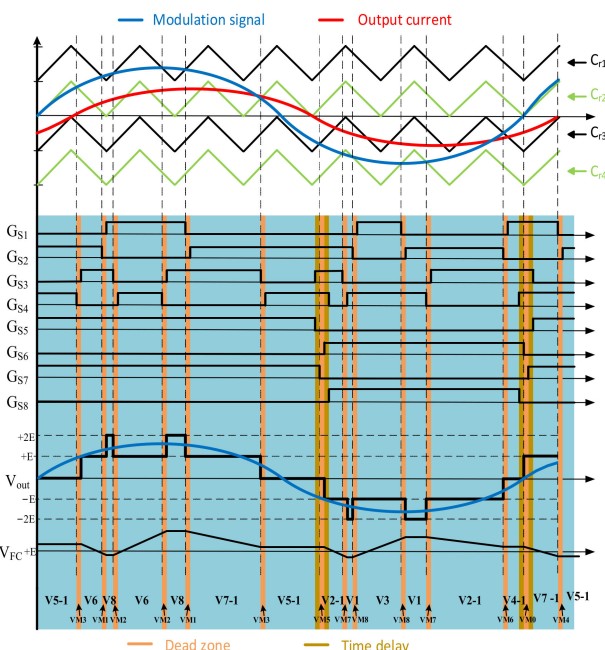

**Figure 12.** The waveform of the floating capacitor voltage.

B. Commutation

The inverter should always be in the state of safety switch during the commutation process, so the problems of shoot-through and high voltage stress during state cutover should be avoided. To distinguish positive current flow from negative current flow, the circuit can be separated into two parts to comprehend the state commutation.

1.  V8 to V6: As shown in Figure 13a, S1 is turned off and after the turn-off delay S2 is turned on. Moreover, with the positive phase current, the state change, current commutation and switching loss occurs at S1 OFF. In contrast, with the negative phase current, the commutation of current and switching loss occurs at S2 ON;

2.  V8 to V7-1: As shown in Figure 13b, it is inevitable to turn off S3 and turn on S4. If the phase current is positive, the commutation of current and switching loss occurs at S3 OFF;

3.  V6 to V5-1: As shown in Figure 13c, it is inevitable to turn off S3 and turn on S4, and if the phase current is positive, the commutation of current and switching loss occurs at S3 OFF;

4.  V5-1 to V7-1: As shown in Figure 13d, it is inevitable to turn off S2 and turn on S1;

5.  V5-1 to V2-1: As shown in Figure 13e, it is inevitable to turn off S5, S7, and S4 and turn on S3, S6, and S8 in the cutover from V5-1 to V2-1. If the phase current is negative, the middle state VM5 generates zero voltage level, not −E voltage level.

Moreover, as shown in Figure 12, under the conduction of the state machine, the switching frequency of switch S5 to S8 is the same as modulation frequency, while switching frequency of switch S1 to S4 is the same as carrier frequency. Moreover, the voltage stress of power devices can be limited to Vdc/4. To sum up, shoot-through problems and overvoltage issues will not be caused by the commutation of current and state cutover. Accordingly, the modulation scheme which is proposed in this section is appropriate for the ANPC-5L inverter.

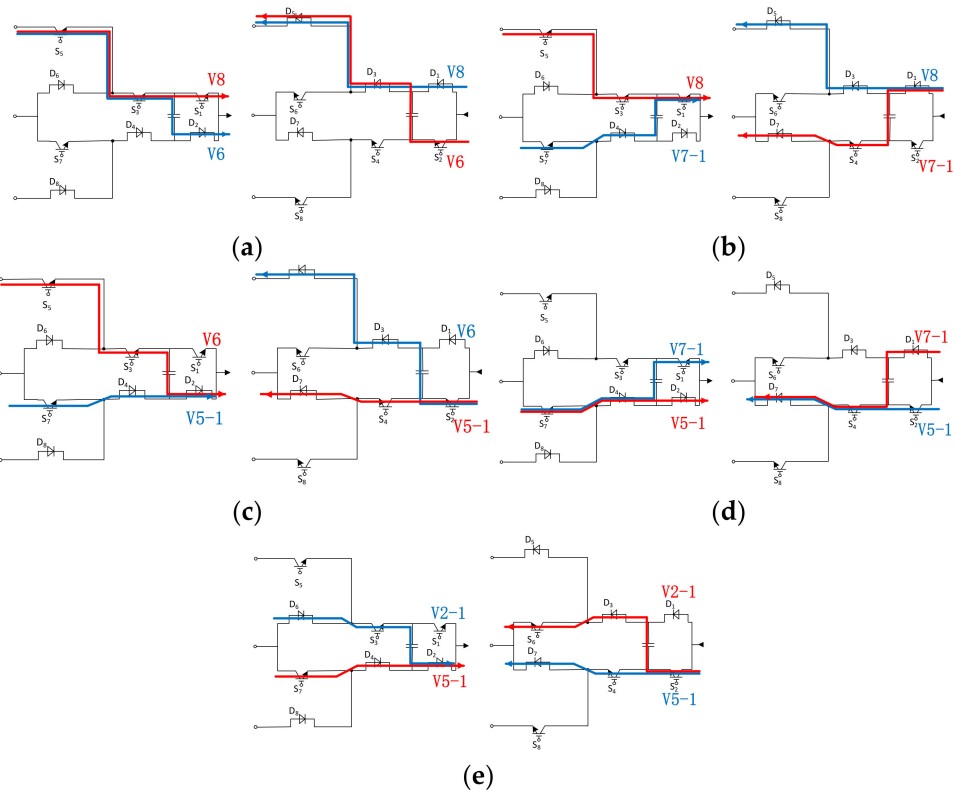

**Figure 13.** State cutover and current commutation. (**a**) V8 to V6. (**b**) V8 to V7-1. (**c**) V6 to V5-1. (**d**) V5-1 to V7-1. (**e**) V5-1 to V2-1.

## 5. Proposed Control Method for Soft Start-Up

The overvoltage issue of the ANPC-5L inverter includes two kinds of problems causing high voltage stress. The first kind is caused by the dead zone of the output voltage switching stage of the half-bridge circuit in series, and the second kind is caused by the soft start process. In the traditional inverter's soft start-up scheme, the dynamic change in voltage may lead to the overvoltage of low-voltage devices.

Start-up is an indispensable process in the control of the photovoltaic inverters, especially among power converters with flying capacitors. There will be very large current stresses on DC-bus capacitors, flying capacitors, and voltage stresses on power switches during the buildup of capacitor voltage if the procedure is not well controlled. Connecting current-limiting resistance in series with voltage sources can limit these stresses in conventional ways. In addition, when bus capacitors are pre-charged but flying capacitors are not fully charged, the voltage stress of several switches will increase by using normal working states, as shown in Figure 14.

Motor windings are used as part of a boost circuit to build up the voltage of flying capacitors with a constant pre-charging current for ANPC-5L converters, and the voltage stress of several switches will double. A pre-charge method applied to flying capacitor multilevel inverters is proposed in [26]; however, it requires plenty of AC contactors and even low-voltage DC power supply, which is not suitable for photovoltaic application.

It is clear that further efforts need to be made to reduce the voltage stress of power devices in the process of soft start-up, as well as to produce flying capacitor pre-charging means with less additional auxiliary circuits. The following part presents an analysis and proposes methods to settle these challenges.

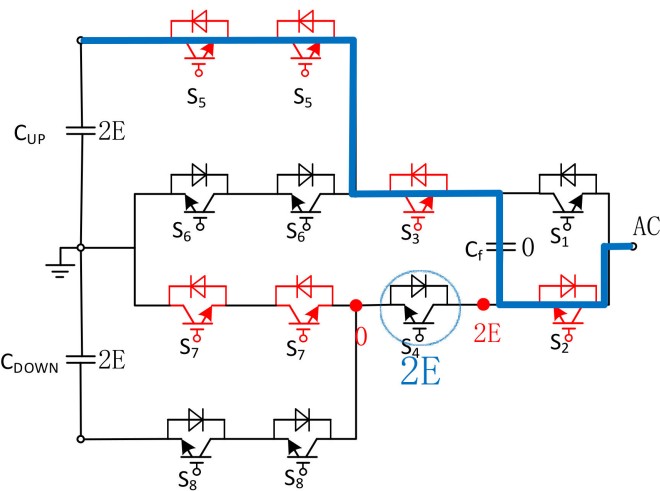

**Figure 14.** High voltage stress without optimal control.

As shown in Figure 15, there are twelve devices in each phase and switches S3, S4, S5 and S8 are used to connect $C_F$ with $C_{UP}$ and $C_{DOWN}$ in parallel. Then, these capacitors can be charged by the DC-link voltage source in the meantime. Assuming the DC-link voltage source is constant, by controlling the main switches S3, S4, S5 and S8, the voltage of the flying capacitor $C_F$ takes priority over the voltage of $C_{UP}$ and $C_{DOWN}$, reaching its reference voltage to ensure that the voltage stress of the switches is not higher than $V_{DC}/4$.

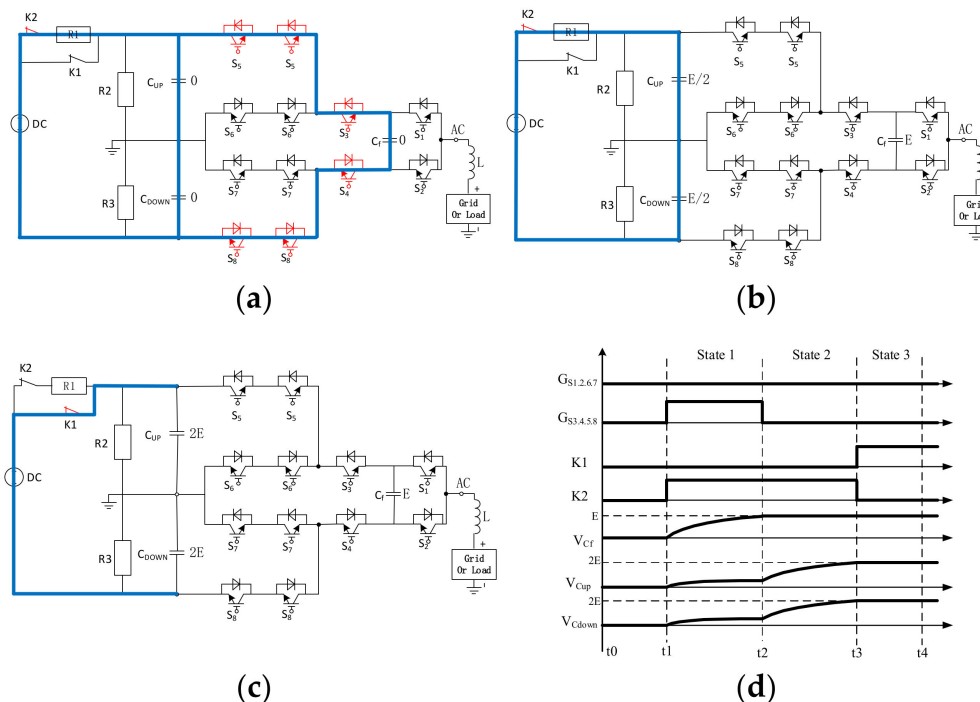

**Figure 15.** Proposed soft start-up process. (**a**) State 1. (**b**) State 2. (**c**) State 3. (**d**) Sequence diagram.

In a summary, the proposed soft start-up method can be divided into three states, as follows:

- State 1: When the upper bus capacitance $C_{UP}$ and the lower bus capacitance $C_{DOWN}$ are zero, and the flying capacitor $C_f$ is not charged and the contactors K1 and K2 are disconnected, the ANPC five-level single-phase converter is in the initial condition with no energy in the capacitors. As shown in Figure 15a, since the main switches S3, S4, S5 and S8 are turned on and the contactor K2 is connected, the DC-link voltage

source charges the upper bus capacitor $C_{UP}$, the lower bus capacitor $C_{DOWN}$, and the flying capacitor $C_f$ simultaneously through the current limiting resistance R1. The voltage-divider resistances, R2 and R3, are placed in parallel with each bus capacitor to avoid the influence of the unbalanced characteristics of the upper and lower bus capacitors. In Figure 15d, the voltage of the flying capacitor $C_f$ and bus capacitors increases gradually from t1 to t2;

- State 2: Until the voltage of $C_f$ arrives at E, a quarter of the total bus voltage and power devices S3, S4, S5 and S8 are turned off and the voltage of bus capacitors will be half of flying capacitor voltage. Thus, the voltage stress of power switches is not higher than $V_{DC}/4$ in the entire stage, which meets its request. As shown in Figure 15b, the DC-link capacitors are going to be charged by the DC-link voltage source at the same time with the voltage divider resistances in parallel with each bus capacitor to avoid the influence of the unbalanced characteristics of the bus capacitors;
- State 3: The voltage of $C_{UP}$ and $C_{DOWN}$ increases gradually until they reach their reference value 2E, as shown in Figure 15d from t2 to t3. When $C_{UP}$ and $C_{DOWN}$ are fully charged, the contactor K1 is connected and K2 is disconnected. Then, the ANPC five-level single-phase converter starts up well with enough energy in the capacitors, as shown in Figure 15c and is ready to work.

The proposed method is applicable to the single-phase of the ANPC-5L inverter and pre-charges through the original PV DC voltage source in the photovoltaic application. The power resistance R1 has an impact on the charge current of the whole startup process, but will not influence the final capacitor voltage. While the converter is in H-bridge topology or in three phase topology, it is still sufficiently practical for the tolerance of the voltage stresses. This method uses the DC side power supply to charge the capacitor in the converter, and has the advantages of simple structure, convenient control, fewer additional auxiliary branches, and reliable soft start-up.

## 6. Experimental Results

To further verify the feasibility of the proposed strategy and theoretical analysis, a low-power prototype was established in the laboratory. Table 3 lists the electrical parameters of the prototype.

**Table 3.** Electrical parameters of the prototype.

| Parameters | Values |
| --- | --- |
| Inverter DC-bus voltage | 400 V |
| Output frequency | 50 Hz |
| Converter rating | 1 kVA |
| Switching frequency | 10 kHz |
| Inductance of filter | 0.3 mH |
| Capacitance of DC-link capacitor | 2 mF |
| Capacitance of flying capacitor | 660 μF |

Figure 16 shows the proposed control scheme for the soft start-up and modulation scheme. $V_f$ is sensed for the balance of the flying capacitor voltage, and $i_L$ is sensed to judge the output current direction. Meanwhile, $V_f$ is sensed for the soft start-up of the inverter. The proposed control scheme can be digitally realized in DSP or CPLD collaborative controllers.

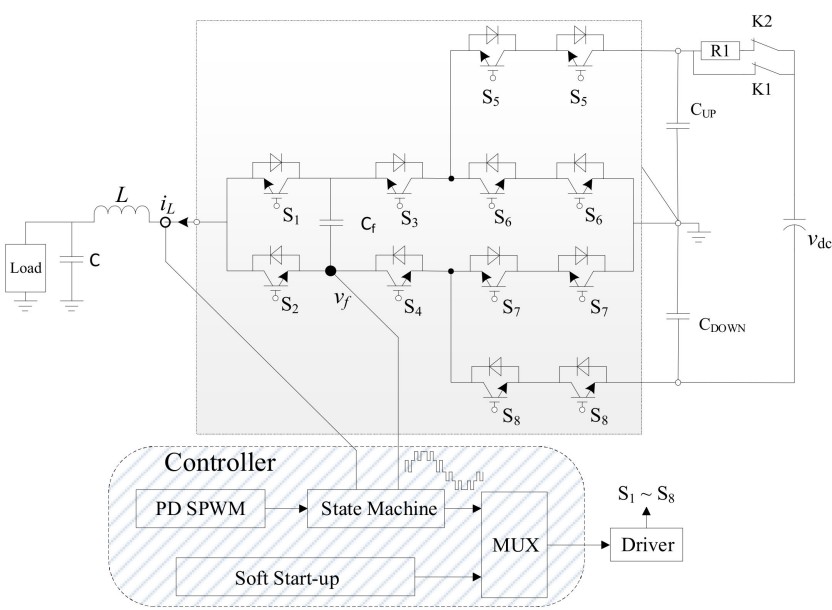

**Figure 16.** Proposed control scheme for soft start-up and modulation scheme.

The voltage of the DC-bus is 400 V, output voltage is 100 Vac and the switching device is 650 V MOSFET IPW60R190Z. As shown in Figure 17, under the traditional modulation strategy, there exists an overvoltage issue in the dead time at the zero crossing point of the waveform. Under the 400 V DC-bus, the voltage stress is about 300 V due to voltage ringing, and if the device is used under a normal bus of 800 V DC, there will be a risk of breakdown.

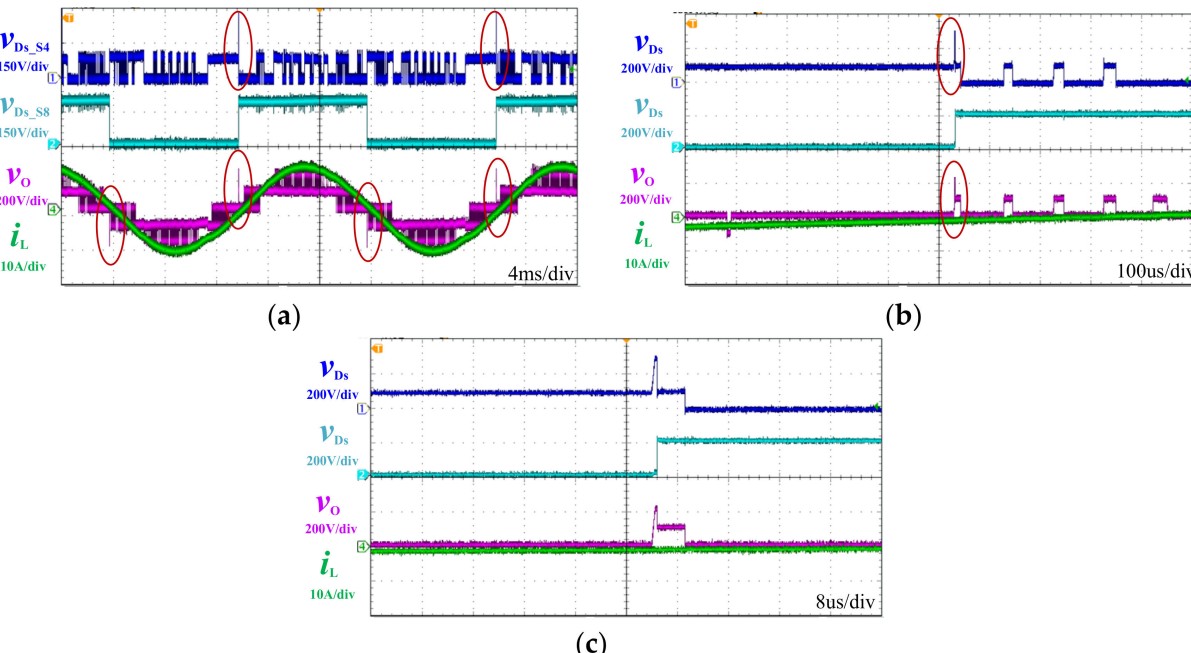

**Figure 17.** The overvoltage stress issue. (**a**) Under fundamental frequency. (**b**) Under multiple switching cycles. (**c**) Under single switching cycle.

To solve the overvoltage stress issue, the driving signal shown in Figure 18 is applied. As shown in Figure 18, there will be no overvoltage stress or output voltage level jump during the same switching process.

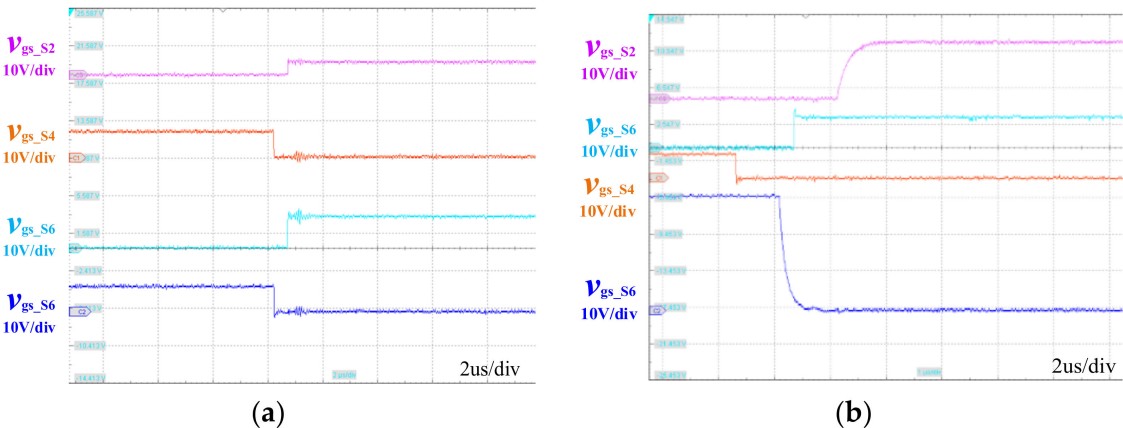

**Figure 18.** (**a**) Traditional and (**b**) modified modulation strategy.

As shown in Figure 19, under the proposed modified modulation strategy, there is no overvoltage issue in the dead time at zero crossing point of the waveform.

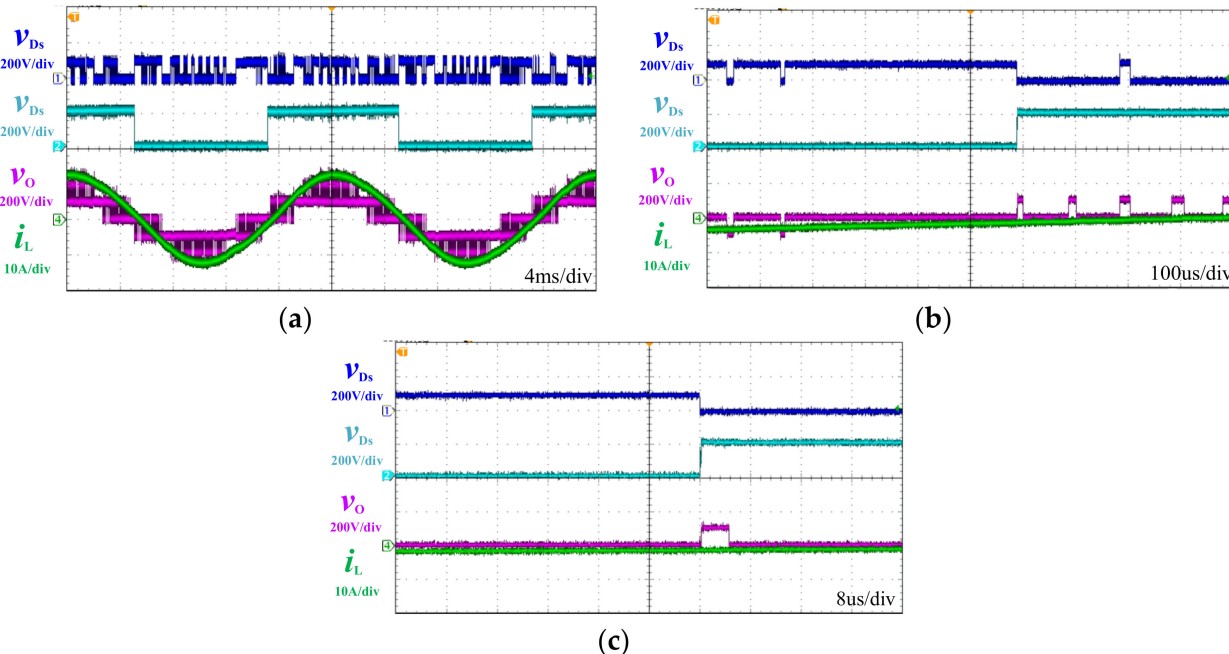

**Figure 19.** The waveform under the proposed modified modulation strategy. (**a**) Under fundamental frequency. (**b**) Under multiple switching cycles. (**c**) Under single switching cycle.

Figure 20 shows the experimental waveforms of Vc_up, Vc_down and VFc during the soft start-up of the inverter under different charging resistances. As discussed in Section 5, the flying capacitor and DC-link capacitor are charged by the DC-link voltage source in the meantime until the flying capacitor reaches its reference voltage Vdc/4 and the DC-link capacitor is charged to the bus voltage continually by controlling several main switches.

The experimental results have demonstrated that, by using the proposed method, the switching frequency of S5–S8 is the same as the fundamental frequency while the switching frequency of the other switches will switch at carrier frequency. Moreover, the stress of power devices can be no more than Vdc/4. By controlling the main switches S3, S4, S5 and S8, the flying capacitor and bus capacitor are fully charged without the overvoltage problem of the switches.

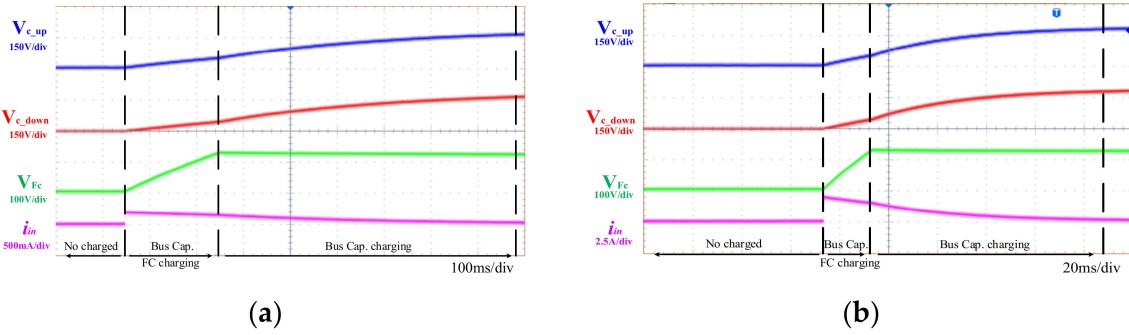

**Figure 20.** Soft start-up under different charging resistance. (**a**) R1 = 500 Ohms; (**b**) R1 = 50 Ohms.

## 7. Conclusions

For the ANPC-5L inverter, the traditional modulation strategy has the security risk of the overvoltage of the power device in the switching dead time when the output current is inductive, which affects the commutation safety and leads to an overvoltage issue. In this paper, the overvoltage mechanism is deduced through the voltage stress analysis of different switching states. Meanwhile, a modified modulation strategy is proposed to solve this issue. In comparison with other modulations, this method provides several free degrees which are used to ensure the elimination of the voltage stress of power devices by choosing favorable circuit states and controlling current commutations. Additionally, it can realize the flying capacitor voltage balance in several carrier wave periods. The implementation of the proposed strategy in the digital system is rather simple. Meanwhile, a novel soft start-up method adapted to the ANPC-5L inverter is also proposed. In addition, an experimental prototype is also built to verify the issue of traditional modulation strategy and the validity and feasibility of the proposed modulation strategy.

**Author Contributions:** Conceptualization, G.C.; methodology, J.Y.; software, J.Y.; validation, G.C. and J.Y.; formal analysis, G.C.; investigation, G.C.; resources, G.C.; data curation, G.C.; writing—original draft preparation, G.C.; writing—review and editing, J.Y.; visualization, J.Y.; supervision, G.C.; project administration, G.C.; funding acquisition, G.C. All authors have read and agreed to the published version of the manuscript.

**Funding:** This work was supported by Shanghai Outstanding Academic/Technical Leaders Plan (20XD1430700).

**Conflicts of Interest:** The authors declare no conflict of interest.

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
