# Peer review of "A Modified Modulation Strategy for an Active Neutral-Point-Clamped Five-Level Converter in a 1500 V PV System"

_electronics, doi:10.3390/electronics11152289_

Round 1

Reviewer 1 Report

A Modified Modulation Strategy for Active Neutral-Point Clamped Five-Level Converter in 1500V PV System

There is a novelty in the work. However, there are some changes that need to be made to reveal this situation. This situation should be compared with the literature and put there.

Reviews:

In Line 92, It should be provided short and informative text about mentioned modulation techniques before comparing methods.
Abbreviations should be used in the same way throughout the text. For example, 0E is used for zero in line 107, while 0 is used alone for zero in line 113.
At which points are the values of VDs_S3, VDs_S8, Vo and IL whose graphs are presented? What is meant by IL? Is it the load current? So where is the load? The points where these potential differences or currents are measured should be shown on a figure.
The literature study is very poor. Studies on this subject in the literature should be mentioned. What methods have been proposed in the literature regarding the overvoltage problem? What are the advantages and disadvantages of the proposed methods? What kind of shortcomings  that make you study about this work did you see in the literature? This situation needs to be revealed in the introduction part.
In addition,in the conclusion part, the contribution to the literature should be presented, if possible, by comparing it with previous studies.
The soft start issue breaks the flow of the text. The connection between these two issues is not well established.

Reviewer 2 Report

The article refers to important problem of right control of 5-level inverter. However, the description of the proposed control method is imprecise. The following notes may help the Authors make the necessary corrections:

1.        Row 115: How to interpret the following entry: (S1-S8:01000000)?

2.        Equation 2: There are same numbers in the parasitic voltages of the switches.

3.        Row 134: Lack of an explanation of the abbreviation: KVL.

4.        Row 142: The state of transistors is not consistent with Figure 8a)

5.        Row 146: The state of transistors is not consistent with Figure 9a)

6.        Figure 12: Incorrectly shown relationship between modulating and carrier signals and transistors driving intervals.

7.        Figure 12: The figure should show a quasi-steady state, with the same waveforms at the beginning of two consecutive positive half-periods of the output voltage. Why is there no periodicity in the waveforms illustrating the states of the transistors?

8.        Row 181: Imprecise letter markings: V7 and V2, not corresponding to Figure 12 and Table 2. What do these markings refer to?

9.        Rows 185 and 186: The content of the opinion is inconsistent with Figure 12.

10.     Rows 187 and 188: On the basis of what such a claim? Intuitively, the opposite is true: the shorter the carrier wave period in relation to the period of the fundamental component, the better the regulation of the capacitor charging voltage.

11.     Row 200: Invalid drawing number

12.     Row 260: There is no drawing showing the time intervals t1 and t2.

13.     Rows 318 to 327: The presented conclusions do not show the specific effects achieved in the Author’s research.

Round 2

Reviewer 1 Report

In the last uploaded study, the criticisms were handled carefully and meticulously. Thus, the fluency and intelligibility of the study increased during reading. The literature review has been expanded with appropriate studies.

The article is suitable to be published.